# Comparing a Sensor for Movement Assessment with Traditional Physiotherapeutic Assessment Methods in Patients after Knee Surgery—A Method Comparison and Reproducibility Study

**DOI:** 10.3390/ijerph192416581

**Published:** 2022-12-09

**Authors:** Jennifer Eymann, Werner Vach, Luis Fischer, Marcel Jakob, Andreas Gösele

**Affiliations:** 1Crossklinik AG, 4054 Basel, Switzerland; 2Basel Academy for Quality and Research in Medicine, 4051 Basel, Switzerland; 3Department of Environmental Sciences, University of Basel, 4056 Basel, Switzerland; 4Crossphysio, 4054 Basel, Switzerland; 5Medical Faculty, University of Basel, 4056 Basel, Switzerland

**Keywords:** wearable sensor, movement exercises, assessment, agreement, reproducibility

## Abstract

Wearable sensors offer the opportunity for patients to perform a self-assessment of their function with respect to a variety of movement exercises. Corresponding commercial products have the potential to change the communication between patients and physiotherapists during the recovery process. Even if they turn out to be user-friendly, there remains the question to what degree the numerical results are reliable and comparable with those obtained by assessment methods traditionally used. To address this question for one specific recently developed and commercially available sensor, a method comparison study was performed. The sensor-based assessment of eight movement parameters was compared with an assessment of the same parameters based on test procedures traditionally used. Thirty-three patients recovering after arthroscopic knee surgery participated in the study. The whole assessment procedure was repeated. Reproducibility and agreement were quantified by the intra class correlation coefficient. The height of a one-leg vertical jump and the number of side hops showed high agreement between the two modalities and high reproducibility (ICC > 0.85). Due to differences in the set-up of the assessment, agreement could not be achieved for three mobility parameters, but even the correlation was only fair (r < 0.5). Knee stability showed poor agreement. Consequently, the use of the sensor can currently only be recommended for selected parameters. The variation in degree of agreement and reproducibility across different parameters clearly indicate the need for developing corresponding guidance for each new sensor put onto the market.

## 1. Introduction

During the last decades, wearable sensors have been developed for a wide range of medical applications [1,2,3]. The underlying technologies have reached a level of maturity allowing them to move from the laboratory and clinical research to a discussion of their application in clinical practice [4,5,6,7]. However, recent systematic reviews still conclude a further need to investigate the reliability and validity of measurements from wearable sensors in clinically relevant populations under routine conditions [8,9].

One specific patient group of interest are orthopedic patients suffering from musculo-skeletal disorders. In these patients the restoration of function is an important goal of any intervention. Assessment of function is often based on clinical examination and corresponding biomechanical tests. If function is related to specific movement patterns, wearable sensors are promising to allow a simple and objective assessment and may replace existing biomechanical assessment methods [9,10]. Such sensors are becoming increasingly commercially available. Even if they turn out to be user-friendly, they face clinicians and patients with the question to which degree the numerical results produced by the sensors are reliable and comparable with those obtained by assessment methods traditionally used.

Recently, Prill et al. [11] presented a first systematic review of commercially available wearable sensors to be used in knee joint rehabilitation. Six sensors could be identified, and the authors concluded a sufficient accuracy in comparison to gold-standard assessment procedures. However, all these sensors had to be coupled with external analytical software for data processing. This limits their clinical usefulness and there is a need for integrated systems. Such a combination of sensors with an analytical engine is given by the “Orthelligent Pro” system released by the medical device company OPED (OPED GmbH, Medizinpark 1, D-83626 Valley/Oberlaindern, Germany). The sensor—to be attached to the knee of a subject—offers the opportunity to assess several parameters related to movement exercises in the lower extremities. All these parameters are of relevance in the rehabilitation process of orthopedic patients. An accompanying app supports the administration of exercises and the visual inspection of the results. The output from the app consist of a single number per exercise and can be tracked over time in the case of repeated applications. The system is (also) intended to be used by patients at home supporting various forms of self-assessment. This way, it can also become part of the communication and interaction between the physiotherapist and the patient. In particular, patients may just present the results of their home assessments to the physiotherapist instead of performing the movement exercise under the supervision of the physiotherapist. In this situation, the physiotherapist must be able to judge the reliability and validity of the output produced by the sensor compared to the assessment methods previously used in his or her routine. The same holds for clinicians used to interpret results of assessments performed by physiotherapists today.

The validity of the sensor has been investigated in a small-scale validation study performing a comparison with gold-standard values produced by a marker-based multi-camera video motion analysis [12]. However, validity only implies that the values of the movement parameters measured by the sensor agree with the values obtained by sophisticated laboratory measurement procedures. This does not imply that they coincide with the values obtained by assessment procedures traditionally used in clinical practice and reflecting current standards. Some care is necessary, as assessment procedures are often labeled with reference to a certain construct (e.g., active flexion, angle reproduction), which can be assessed using different parameters or different procedures to measure the parameter of interest. Hence, also for a sensor with proven validity, (systematic) discrepancies between the values produced by the sensor and values obtained as part of a traditional assessment cannot be excluded. Knowledge about such differences is essential to ensure that results based on the sensor are correctly interpreted.

The aim of the current study is to investigate the magnitude of differences in numerical results between a sensor-based assessment with a specific, commercially available sensor and a traditional assessment following current standards. Information about the magnitude can inform potential users of this sensor with respect to its role in replacing traditional assessment techniques. In addition, the reproducibility of the measurements both based on the sensor and based on traditional assessments will be investigated, as low reproducibility may be a source for lack of agreement [13]. The reporting of the study follows the GRRAS guideline [14].

## 2. Materials and Methods

### 2.1. Definition of Assessment Procedures

The “Orthelligent Pro” system allows the measurement of exercise-specific movement parameters of the lower extremities in the knee joint area. It is based on a 9-axis inertial measurement unit (IMU) collecting data at a frequency of 200 Hz. It combines the functionality of an accelerometer, a rotation rate sensor (gyroscope), and a magnetometer (compass).

The system included detailed instructions for thirteen different exercises. For each exercise the sensor measures one predefined numerical parameter. Eight of the parameters were selected for this study due to the possibility to measure the same (or a conceptually similar) parameter based on a traditional assessment. For the traditional assessment, procedures were chosen that are described in guidelines and textbooks and are widely used in clinical practice [15,16,17,18,19,20,21,22]. They as based on varying technologies such as the universal goniometer, visual assessment, or force plate.

Table 1 describes how the exercises were performed in the sensor-based and the traditional assessment and how the parameter of interest was assessed. Videos illustrating the sensor-based assessment procedures are available at the website of the manufacturer (https://oped.de/produkte/orthelligent-pro accessed on 14 September 2022). Table 2 summarizes the parameters finally measured and compared between the two modalities (i.e., sensor-based and traditional assessment) within this study.

Although the aim was to align the traditional assessment to the sensor-based assessment, some (conceptual) differences could not be avoided. For three exercises the performance during the sensor-based assessment was video-recorded and the traditional assessment was based on inspection of this video. However, for two of these exercises (*One leg squat* and *Drop jump*), the traditional assessment does not include a numerical quantification, but only a scoring on an ordinal scale. Only for *Side hops*, a quantification similar to the sensor-based assessment was possible. Three exercises assessed mobility: *Passive flexion*, *Passive extension* and *Active flexion*. Here, the traditional assessment was based on repeating the exercise with a different set up: these assessments are traditionally performed with the subject lying in a supine position, whereas the Orthelligent Pro system prefers a sitting or standing position. The sensor-based assessment of *Vertical jump* was performed already on a force plate directly allowing the traditional assessment. Similarly, for *Angle reproduction*, the angles were assessed with the universal goniometer in parallel to the sensor-based assessment.

### 2.2. Patient Recruitment and Conduct of Assessment

Patients were recruited from the patient population of the crossklinik, a medical clinic specialized in orthopedics and sports medicine with an interdisciplinary team of physicians, physical therapists, and sports scientists. Patients were eligible for inclusion if they underwent arthroscopic knee surgery between 4 months and 15 months prior to the contact. Patients were excluded if they were below 18 years, underwent other surgery in the lower extremities within the last 4 months, suffered from severe acute pain, had contraindications to perform the exercises, underwent surgery also at the contralateral leg since the arthroscopic knee surgery or could not follow the instructions due to language or cognitive problems.

Eligible patients were identified in the patient administration system. They were contacted by phone. They were informed about the study and asked for participation. In case of interest, they obtained written information on the study and a consent form and were scheduled for a visit. At their visit, the consent form was checked. The assessment was performed at the biomechanics lab of the crossklinik. The patients wore sports clothing and no shoes. They were offered a 10 min warmup on the ergometer.

The sequence of the exercises was the same in each patient. The order coincides with that shown in Table 1 and Table 2. The order was chosen to optimize the workflow: exercises requiring the same positioning of the patient or the same equipment were grouped together. First, the three traditional mobility assessments were performed in a row, first with the contralateral and then with the injured leg. Then, the eight sensor-based assessments were performed, first all of them with the contralateral and then all of them with the injured leg. The traditional assessments were partially performed in parallel (see Table 1). After finishing all assessments, the whole procedure (expect the warming up) was repeated. After finishing the second round, the results of the sensor-based assessment were discussed with the patient. The video-recordings were evaluated later after the visit of the patient.

The sensor-based assessment followed closely the instructions delivered by the manufacturer. The sensor was placed on the thickest part of the calf (approximately 3 finger lengths below the tibial tuberosity) on the lower leg, as otherwise the holder may tilt, resulting in incorrect measurement results. The app accompanying the sensor was installed on a tablet and used to administrate the assessment. Before each exercise, an instruction video was watched together with the patient. In addition, standardized verbal instructions were given to the patient (see Appendix A). The test administrator informed the subjects about the start and end of the actual measurement. Each measurement period started with a 3 s countdown during which the leg should be kept still, and the sensor is calibrated. During the exercises, the patients could not watch the tablet display, except for the training phase of the *Angle reproduction* exercise.

During the sensor-based assessment, prior to the *Passive flexion* and *Active flexion* exercise, the patients performed one trial attempt to ensure that the movement was performed without evasive action. Prior to *One leg squat* and the three jump-based exercises (*Drop jump*, *Side hops* and *Vertical jump*), the patients performed up to three trial attempts. When the subjects felt confident in the movement process, the actual measurement was performed. The examiner could at any time decide to stop an exercise or to omit an exercise if the exercise was too demanding for the patient. The results from the goniometer-based assessment were immediately recorded on a CRF. The results from the sensor-based assessment and the force-plate based assessment was extracted after the patient visit was finished and recorded on a CRF. The same holds for the video-based assessments.

After finishing the first 20 patients, the *Drop jump* was removed from the test battery. This decision reflects that the sensor indicated a tilt angle of exactly 0 for the vast majority of patients. All assessments were conducted by one examiner (JE). The examiner had a background in sports science (Master of science in sport, exercise, and health—prevention and health promotion) and conducted assessments in the context of professional occupation as a sports scientist at the crossklinik with 2.5 years of professional experience.

### 2.3. Statistical Analysis

Reproducibility and agreement were investigated with respect to the raw measurements. A preprocessing was not performed. Reproducibility and agreement were also investigated with respect to the deviation between the injured and the contralateral leg. Such deviations can be quantified in different ways, e.g., as ratios or differences. Appendix A depicts the relation between the measurement pairs. The expected correlations between the injured and contralateral sides were observed in all exercises, but to varying degrees. Specifically, for *Angle reproduction* and for *One leg squat* assessed by the sensor the correlation was rather low. The shape of the point clouds in the scatter plots does not clearly favor the use of differences or ratios: neither a clear shift nor a clear proportional reduction is observed. In addition, the range of measured values includes 0 for some exercises. In these cases, using a ratio is not feasible. To approach a uniform handling across all exercises, we used the difference to assess the deviation: the value obtained at the injured leg was subtracted from the value at the contralateral leg. Positive values indicate that the measurement was larger at the contralateral leg. We refer to these values in the sequel as side differences.

Reproducibility of the raw values was depicted by scatterplots comparing the measured values from the second round with the values from the first round. It was quantified by the intra class correlation coefficient (ICC). ICC values are comparable across measurement procedures using different measurement scales if the same population is analyzed. In addition, we report the mean and standard deviation of the differences, the *p*-value of a paired t-test testing the null hypothesis of a mean difference of 0, the limits of agreement, the mean absolute difference and the root mean square error. Except the *p*-value, these quantities are not comparable across different measurement scales. Reproducibility of the side differences was analyzed analogously comparing the computed values from the second round with the computed values from the first round.

Agreement of the raw values and the side differences, respectively, between the two modalities was analyzed in a corresponding manner. However, for *One leg squat* and *Drop jump* only the association between the two modalities can be considered because the two modalities used different scales. The Pearson correlation coefficient was used to assess the degree of association.

In verbalizing the degree of reproducibility or agreement, we made use of the following classification of ICC values: “high” for values above 0.85; “moderate” for values above 0.7; “fair” for values between 0.4 and 0.7 and “poor” for values below 0.4.

The intended sample size for this study was 40. With this sample size, an ICC of 0.7 can be estimated with a precision (expected half-length of the 95% confidence interval) of 0.161 and an ICC of 0.85 with a precision of 0.088 [25,26,27]. It is recommended to include in method comparison studies a wide and realistic range of values for the variables investigated [28]. This recommendation is approached by imposing no restrictions on the age and current functional status of the patients.

### 2.4. Ethical Approval and Registration

The study was approved by the Ethikkommission Nordwest- und Zentralschweiz (EKNZ) under reference number 2021-00332. The study was conducted in accordance with the Declaration of Helsinki and the guidelines for Good Clinical Practice (GCP). The study is registered at clinicaltrials.gov under the registration number NCT04939389.

## 3. Results

### 3.1. Population and Assessment Procedure Characteristics

Overall, 126 patients were contacted and 36 agreed on participation, showed up at the scheduled visit, and gave informed consent. The assessments were performed between 26 January 2022 and 28 March 2022. Three patients started the assessment, but the assessment was stopped due to unexpected pain. The data of these patients was not included in the analysis.

Table 3 summarizes basic characteristics of the patient population. The population was rather balanced with respect to gender and the side of the injured leg. Partial meniscectomy and ACL reconstruction were the most frequent types of surgery. 9 patients reported health problems related to the leg on the contralateral side. Age ranged from 20 to 70 years with an average of 43 years. The BMI was on average 25, and five patients had a BMI above 30. On average patients had surgery 10 months before the assessment.

Table 4 summarizes basic characteristic of the distribution of the parameters assessed by the two modalities in each exercise. In a few cases, not all exercises could be performed. This was due to pain, anxiety, or lack of strength. In the case of *Vertical jump*, it once happened that the force plate did not recognize the jump as such due to poor jump quality, but the sensor did. The range of the observed values differed substantially between the two modalities with respect to the angles reported for *Passive flexion*, *Passive extension*, and *Active flexion*. A comparison of mean values between the injured and the contralateral side indicated more favourable values on the contralateral side for *Passive flexion*, *Active flexion*, *Drop jump*, *Side hops*, and *Vertical jump*.

### 3.2. Reproducibility of Raw Measurements

Figure 1, Table 5 (columns 1 and 2), and Appendix A depict the reproducibility of the raw measurements for both modalities. The ICCs indicate high reproducibility for *Passive flexion*, *Active flexion*, *Side hops* and *Vertical jump* for both modalities and for *Passive extension* and *Drop jump* for the traditional assessment. The latter two show a reduced reproducibility for the sensor-based assessment. For both modalities, the reproducibility of *One lag squat* is fair and the reproducibility of *Angle reproduction* is poor. For both exercises, the reproducibility is lower in the sensor-based assessment than in the traditional assessment. Further results presented in Appendix A indicate a learning effect for *Side hops* under both modalities.

### 3.3. Reproducibility of Side Differences

Figure 2, Table 5 (columns 3 and 4), and Appendix A depict the reproducibility of the side differences between the injured and the contralateral injured leg for both modalities. Only for the traditional assessment of *Passive Extension* a high reproducibility can be observed—the sensor-based assessment is distinctly less reproducible here. Moderate reproducibility is observed for *Vertical jump*, and fair reproducibility for *Passive flexion*, *Active flexion* and *Side hops*, whereas the reproducibility is poor for *Drop jump*, *One leg squat* and *Angle reproduction*—always with a very similar magnitude for the two modalities.

### 3.4. Agreement of Raw Measurements between the Two Modalities

Figure 3, Table 6 (left column), and Appendix A depict the agreement of the raw measurements between the two modalities. High agreement can be observed for *Side hops*, *Vertical jump*, and *Angle reproduction*. As pointed out above, *Passive flexion*, *Passive extension*, and *Active flexion* suffer from differences in the range of possible values between the two modalities. Consequently, the agreement is poor. However, even when this is taken into account by considering association instead of agreement, correlations in the magnitude of only 0.3 to 0.5 are observed, which are distinctly lower than the correlations observed for *Side hops*, *Vertical jump*, and *Angle reproduction*. Low correlations were also observed for *One leg squat* and *Drop jump*, for which ICC values could not be computed.

### 3.5. Agreement of Side Differences between the Two Modalities

Figure 4, Table 6 (right column), and Appendix A depict the agreement of the computed side differences between the two modalities. For *Side hops*, *Vertical jump*, and *Angle reproduction,* a moderate-to-high degree of agreement can be observed. Despite the poor agreement in the raw values, *Passive flexion*, *Passive extension* and *Active flexion* could in principle show a perfect agreement in the side differences, as the differences in the measurement ranges can cancel out. However, this is not the case: they show only a poor to fair degree of agreement. Finally, side differences correlated only to a low degree when considering *One leg squat* and *Drop Jump*. For *Drop jump*, at least some agreement with respect to the absence of a side difference is observed in Figure 4.

## 4. Discussion

### 4.1. Summary of Results

Sensor-based assessments and traditional assessments can lead to nearly the same values, i.e., they show a very high agreement of the raw measurements. In our case, this was the case for *Side hops*, *Vertical jump*, and *Angle reproduction*. However, this was not the case for the other exercises considered.

### 4.2. Side Hops

At first side, it may be not surprising that there was a high agreement for *Side hops*, as a single side hop is a movement which should be easily detected by an IMU. However, there are some challenges with side hops in clinical routine: side hops may be executed in an erroneous fashion, such that they should not be counted. This may be due to an incorrect choice of the distance, which has to be chosen for each individual, temporary loss of balance, or lack of power or of confidence in the injured leg. Counting and identifying these incorrect hops can be challenging for the examiner, in particular if fast, short hops are executed. Unfortunately, the sensor does not offer a solution to this problem. Since the same count of erroneous hops was used for both modalities, this issue could not affect agreement in our investigation.

However, counting the overall number of hops can already be challenging if patients perform fast, short hops. Indeed, differences of more than 10 hops were observed in a few patients. According to our experience, this happened exactly in patients with a somewhat questionable quality of their hops. As these discrepancies can appear independent of the leg involved, this may explain the moderate agreement with respect to side differences. The reproducibility of *Side hops* was high for the raw measurements but only moderate for the side differences. This pattern has previously also been reported in investigating several smart devices [29].

### 4.3. Vertical Jump

The high agreement with respect to *Vertical jump* is a non-trivial result, as the sensor is based on a completely different measurement approach (assessing the spatial movement of a point) compared to the traditional assessment (translating a force time history). The manufacturer of the sensor informed us that differences up to 10 cm are possible. Similar to the side hops, there can be also issues related to the quality of the jumps. Nevertheless, with one exception, there was never a difference larger than 3.1 cm for the raw measurements and of 3.6 cm for the side difference. With respect to the reproducibility of the raw measurements, a high reproducibility was observed for both modalities, which is in line with a previous investigation [30].

### 4.4. Angle Reproduction

The high agreement with respect to *Angle reproduction* simply indicates that the sensor has the ability to measure a knee angle with the same precision as the universal goniometer, because the knee positions were identical under the two modalities. This example also illustrates that a high agreement between the modalities is not an indication for the clinical utility of a parameter: The reproducibility of *Angle reproduction* was poor under both modalities.

At first sight, this may be due the decision to reproduce fully the sensor-based assessment including the random choice of the angle to be reproduced. However, if angle reproduction is a useful concept to evaluate the knee status of a patient, it should be independent of the angle chosen. Hence, this decision is not unreasonable. In addition, differences in the angle to be reproduced only partly explain the low reproducibility. If only those measurement pairs with a difference of maximally 10° in the angle to be reproduced are used (allowing 31 pairs to be included), the ICC-values depicting the reproducibility of raw values increase only from 0.27 to 0.50 (sensor-based assessment) and from 0.34 to 0.48 (traditional assessment), i.e., the reproducibility is still only fair.

A comparison between a smartphone-based goniometer and an isokinetic dynamometer as measurement modalities [31] reported very similar results: High agreement, but only fair reproducibility for both modalities.

### 4.5. Mobility Assessments

With respect to *Passive flexion*, *Passive extension*, and *Active flexion*, the differences in the set up seem to have a substantial impact. Not only the observed value ranges differed substantially and made an exact agreement unlikely, but even the correlation between the modalities was only moderate. With respect to *Passive flexion*, in the sensor-based assessment the proband has to move the leg, and not the assessor. In addition, the seated position may allow more evasive movements in the hip and a large abdominal or chest circumference could also prevent the knee from being brought into maximum flexion.

With respect to *Active flexion*, the sensor-based assessment cannot assess the movement of the thigh separately from the movement of the lower leg. Consequently, the setup requires the tight to be kept fixed in contrast to the traditional assessment. In addition, in the sensor-based assessment, the back thigh musculature is challenged against the force of gravity—a further difference to the traditional assessment. Especially after a removal of the semitendinosus tendon (hamstring), there is often a clear deficit here. The reproducibility of the flexion-related parameters was high in our study, which has been previously reported in some, but not in all studies on reproducibility [32]. The possibility to assess flexion angles even by smartphone apps has been demonstrated previously [33,34,35].

With respect to *Passive extension*, the sitting position is a practical challenge in the sensor-based assessment. It is necessary to ensure that the test persons do not sit too far back on the seat. If the back of the thigh also touches the seat instead of just the buttocks, this can change the measurement results. The manufacturer also warns against this. In practice, however, it is very difficult to place the patients exactly in the desired position on the block. Our results indicate that this may be a serious problem: the traditional assessment showed an excellent reproducibility even for side differences, but the sensor-based assessment showed only a moderate reproducibility. Previous studies on the reproducibility of goniometer-based knee extension range measurements have reported ICCs in the magnitude of 0.41 to 0.86 [36].

### 4.6. Knee Instability

Finally, two exercises aimed at catching the instability of the knee joint. The degree of stability of the knee joint is a sensitive predictor of re-injury of the anterior cruciate ligament upon resumption of sports activities, for example, in the case of ACL replacement [37]. In addition to assessing the risk of re-injury, the test is also useful in the prevention of knee and ACL injuries [38,39]. Traditionally, knee stability has been assessed on binary or ordinal scales representing a subjective grading and it has rarely been quantified [40]. The objective measurements based on a sensor may promise here an interesting alternative. However, only a low correlation between the two modalities was observed for both tests, and the reproducibility of the raw values was less pronounced for the sensor-based assessment. Hence, it remains questionable whether the sensor-based assessment offers an advantage.

The visual assessment of knee instability is well known to be a challenging task. Only a moderate reproducibility has been reported for the *Drop jump* based on a single study [41] as well as for the *One leg squat* in a meta-analysis [40]. For a sensor the assessment of knee instability seems to be a challenging task, too. Indeed, the validation study indicated already only a limited validity of these measurements for the sensor investigated in this study [12]. This may be related to the fact that several joints and muscles are involved in the movements performed during a *One leg squat* or a *Drop jump,* and evasive movements may take place in the foot, hip, or torso.

### 4.7. Implications for the Specific Sensor Investigated

For the specific sensor investigated, its use for assessing the number of side hops and the height reached in a vertical jump is supported by our results, as the measurements are highly reproducible and agree with the traditional assessment. However, it should be kept in mind that for both modalities, the reproducibility of side differences is imperfect. The angles produced under *Active flexion* and *Passive flexion* show the same degree of reproducibility in the sensor-based assessment as in the traditional assessment, but the values tend to differ substantially between the two modalities and show only a moderate correlation. Here, there is a need for further research clarifying whether this is associated with a shift in the clinical value of the assessment. For *Passive extension,* the sensor-based assessment is much less reproducible compared to the traditional assessment and probably offers no advantage. The use of the sensor to assess instability cannot be recommended.

The observed variation in agreement and reproducibility across the different parameters underlines the relevance of studies focusing on the utilization of sensors in clinical practice as expressed by several authors [8,9,11].

### 4.8. Implications for the Use of Sensor-Based Assessments in General

In the physiotherapeutic management of patients, moving from a traditional assessment to a sensor-based assessment can be based on different expectations. First of all, we may wish to replace a well-established assessment with a new one allowing to involve directly the patient and to focus the physiotherapeutic management on advice and training rather than assessment. Here, it is essential to ensure that the sensor produces values directly comparable to those of the traditional assessment (or that a valid translation rule is provided) in order to avoid misleading interpretations by the physiotherapist.

However, a sensor-based assessment may also offer an improvement compared to the traditional assessment in the sense to provide more accurate or more relevant values. Instability of the knee may be an example for this, as the traditional assessment procedures based on a visual assessment may constitute an imperfect standard. In this case, it can be an advantage that the sensor-based values differ from those obtained by a traditional assessment. However, it is necessary to perform additional investigations to ensure that the alternative values are also (more) clinically relevant, for example, that they can (better) predict the future course of a patient.

Our investigation demonstrates that side differences are often much less reproducible than raw measurements. A decrease in reproducibility has to be expected, as side differences are affected twice by the measurement error in contrast to raw measurements. However, the magnitude of the differences should be seen as a reminder to interpret the degree of reproducibility of raw measurements not as the degree of reproducibility of side differences.

### 4.9. Limitations of the Study

The study was performed by one assessor with a specific background. This limits the generalizability of the results. On the other hand, it ensures that the results are not affected by inter-observer variation.

The traditional assessment procedures used in physiotherapy are not completely standardized. Other procedures may be used for the same movement parameters, and there may be subtle differences in the application of standard procedures. This limits again the generalizability of the results.

Sensor-based (self-)assessment aims at longitudinal observations allowing not only to study a patient’s current condition, but also the progress in the recovery process. This important aspect could not be investigated in this study.

Although commercially available, the Orthelligent Pro system is still under development and internal measurement processes are subject to change. Hence, it cannot be ensured that the results presented in this paper apply also to future versions of the system. According to the manufacturer, the measurement algorithms for *Drop jump* and *One leg squat* were updated after finishing the measurements of this study.

## 5. Conclusions

Sensor-based assessments offer the opportunity for a self-monitoring of patients after knee surgery and may have the potential to improve the communication between patients and physiotherapists. However, they can only replace traditional assessments if the clinical value of the measurements produced is clarified. The variation in degree of agreement and reproducibility observed in this study across different parameters clearly indicates the need for developing corresponding guidance for each new sensor put onto the market.

## Figures and Tables

**Figure 1 ijerph-19-16581-f001:**
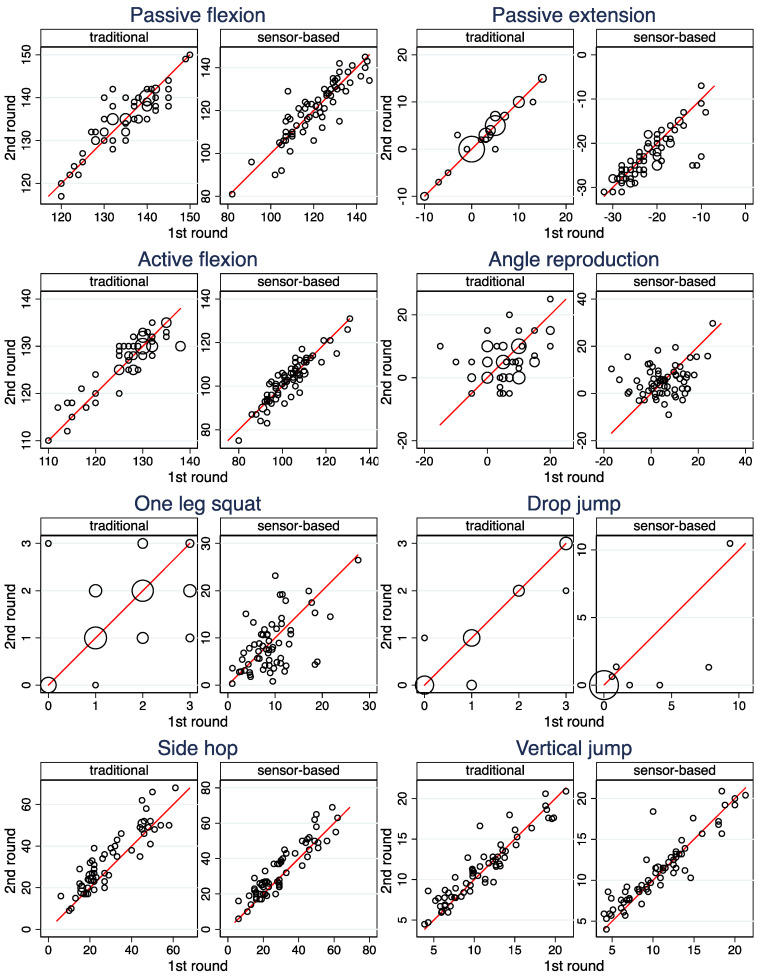
Scatterplots of the measurements made in the second and the first round. Observations from both legs are included in the same scatter plot. The area of the circles is proportional to the number of observations. The red line indicates equality of the two measurements.

**Figure 2 ijerph-19-16581-f002:**
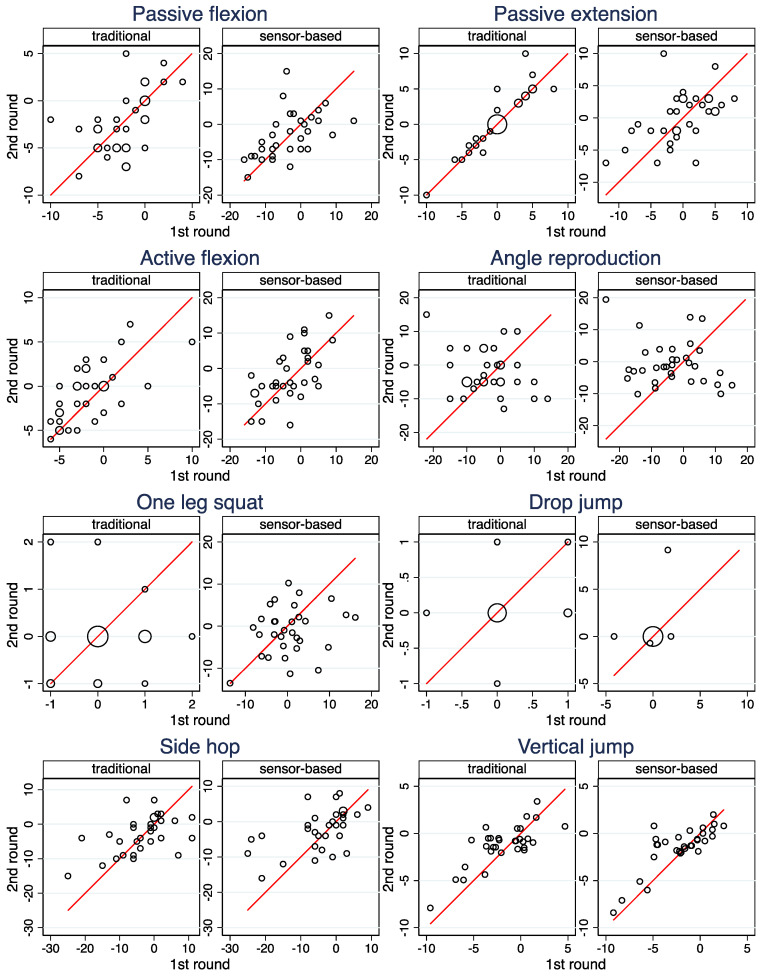
Scatterplots of the computed side differences in the second and the first round. The area of the circles is proportional to the number of observations. The line indicates equality of the two computed values.

**Figure 3 ijerph-19-16581-f003:**
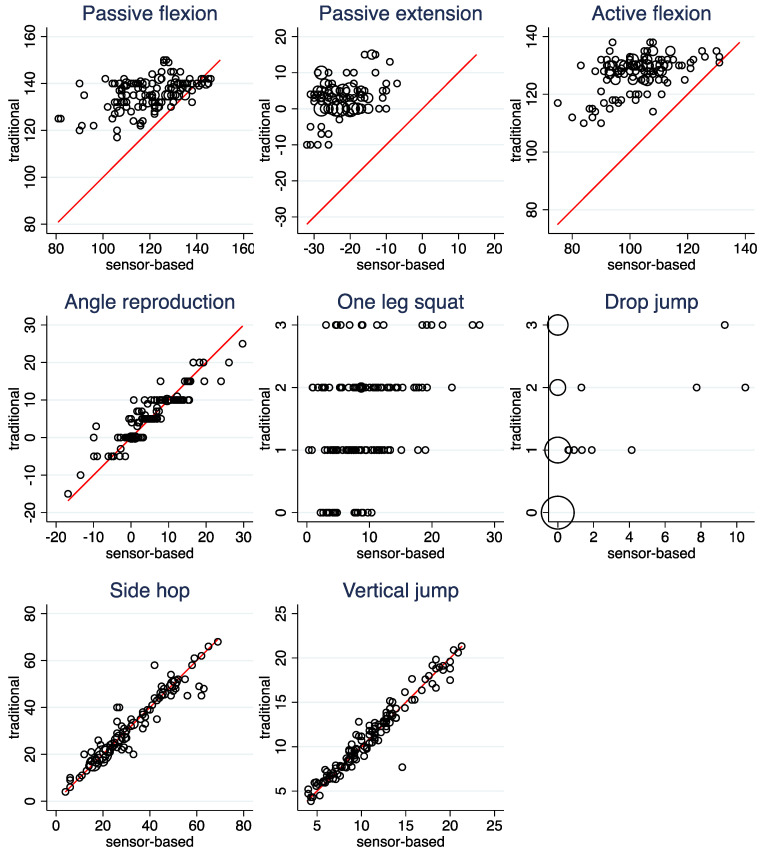
Scatterplots of the raw measurements comparing the sensor-based and the traditional assessment. Observations from both legs are included in the same scatter plot. The area of the circles is proportional to the number of observations. The line indicates equality of the two measurements.

**Figure 4 ijerph-19-16581-f004:**
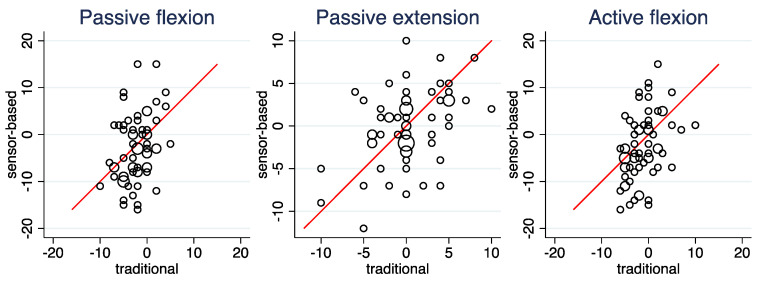
Scatterplots of the computed side differences comparing the sensor-based and the traditional assessment. The area of the circles is proportional to the number of observations. The line indicates equality of the two computed values.

**Table 1 ijerph-19-16581-t001:** Assessment procedures used for the eight exercises.

Passive flexion of the lower leg (knee joint)	
Sensor-based assessment	The subject sits on the floor with legs extended. The leg with the sensor is pulled into maximum knee flexion with the help of the hands (grip behind the back of the knee). The leg is then pushed back into the extended starting position. The movement is performed once. The heel remains in contact with the ground during the entire movement. The sensor measures the maximum flexion in degrees.
Traditional assessment *	The subject lies flat in a supine position on a mat on the floor. The therapist places one hand on the subject’s thigh and the other in the distal area of the lower leg. The therapist guides the knee joint into maximum knee flexion. Once the final position is reached, the extent of movement (180°—angle between the longitudinal axis of the femur and the tibia) is measured with the universal goniometer.
Passive extension	
Sensor-based assessment	The subject sits with a straight back on an elevation of 10 cm. A roll (BLACKROLL STANDARD) with a diameter of 15 cm is placed centrally in the back of the knee of the leg to be tested. After the countdown, the knee is lifted slightly so that the roll can be pulled out. The subject should straighten the knee naturally without force. The heel remains in contact with the ground during the entire movement. The sensor measures the maximum passive extension in degrees.
Traditional assessment *	The extension test is performed as follows: The subject lies flat in the supine position on a mat on the floor. In the final position, the extent of movement (hypomobility: 0°—angle between the longitudinal axis of the thigh and lower leg or hypermobility: 0° + angle between the longitudinal axis of the thigh and lower leg) is measured with the universal goniometer.
Active flexion of the lower leg (knee joint)	
Sensor-based assessment	The subject stands straight with legs extended. The knee is bent to the maximum and returned to the starting position (the end position does not have to be held). This is done slowly and without momentum, the knees remain together. Only the lower leg moves, without the whole leg shifting backward or forward. To ensure this, the subject leans the thighs against the back of a chair. The sensor measures the maximum flexion in degrees.
Traditional assessment *	The subject lies flat in a supine position on a mat on the floor. The subject brings the leg to be examined into knee flexion under his own strength as far as possible. The heel remains in contact with the floor. In the final position, the extent of movement (180°—angle between the longitudinal axis of the thigh and lower leg) is measured with the universal goniometer.
Angle reproduction	
Sensor-based assessment	The app randomly specifies an angle that must be reached by the user. The randomly selected angle will not exceed the value of the angle measurement obtained for Active flexion. The test procedure itself is the same as for the Active flexion. The test procedure is used in a training phase, during which the subject can watch the actual angle achieved on the display. Then, the procedure is repeated 3 times without watching the display. Between each measurement, the subject should walk a few steps. The sensor measures the end angle in degrees for each reproduction attempt. Average deviation over the three trials is calculated by the sensor.
Traditional assessment	During the assessment run with the sensor, the angles are measured with the universal goniometer and rounded to the nearest 5 degrees, taking into account that the leg cannot be held completely still in the angled position. The average deviation over the three tests is noted.
One leg squat	
Sensor-based assessment	An elevation of either 10 cm, 20 cm or 30 cm is used to perform the test. The subject stands on the elevation with an extended leg. After the countdown has elapsed, a one-legged squat is performed. The subject goes into a deep bend until the heel touches the floor. The upper body should remain upright. The height of the elevation must be chosen so that it is challenging for the patient. According to the manufacturer, 20 cm is a good starting value for averagely athletic subjects. The test must be performed exactly once within the time interval. The sensor measures the inward tilt of the lower leg in degrees.
Traditional assessment	The test performance with the sensor is recorded on video from the front with a tablet. Subsequently, knee valgus is assessed using an ordinal scale: 0 = no valgus (central patella always in line with hip joint and second toe) 1 = small valgus (small oscillatory movements around neutral position) 2 = moderate valgus (central patella over great toe) 3 = severe valgus (central patella medial from great toe)
Drop jump	
Sensor-based assessment	A 30 cm elevation is used to perform the test. The subject stands upright on the elevation. The hands are placed on the hips. The subject jumps off with both legs simultaneously and lands about 30 cm in front of the box. After landing, the subject immediately jumps back up. The goal for the subject is to achieve a short ground contact time. The medial drift (the inward tilting) of the knee is measured from the sensor in a time range up to about 0.2 sec after the first landing impulse.
Traditional assessment	The test performance with the sensor is recorded on video from the front with a tablet. Subsequently, knee valgus is assessed using an ordinal scale: 0 = no valgus (central patella always in line with hip joint and second toe) 1 = moderate valgus (over great toe) 2 = severe valgus (medial from great toe)
Side hops (30 s)	
Sensor-based assessment	Before the test, a distance (1 cm, 10 cm, 20 cm, 30 cm or 40 cm) is determined and marked with tape on the floor. The distance must be chosen so that it is challenging for the patient. According to the manufacturer, 20 cm is often a good starting value and optimal for averagely athletic patients. The subject has 30 s to jump as many times as possible with one leg over the given distance. The entire body should be involved in each jump. If the upper body always remains in the same place and only the foot jumps quickly from left to right, the distance is too short. During the measurement, the sensor automatically counts the ground contacts. The examiner counts the errors, such as touching the line or a correction jump.
Traditional assessment	The test performance with the sensor is recorded on video from the front with a tablet. Subsequently, the jumps are counted. The error count is not repeated.
Vertical jump (on one leg)	
Sensor-based assessment	The subject takes up an upright position with the hands placed on the hips. After the countdown has elapsed, a single-leg jump is performed. The jump should be as high as possible. Momentum may be gained from the upper body. The jumping leg remains stretched in the air. The landing is also done with one leg. The movement of the center of gravity from the standing position to the maximum height is determined from the sensor.
Traditional assessment	The sensor-based assessment is already performed on a force plate (MLD Test Evo 2, SP-Sport, Innsbruck, Austria) to measure the ground reaction forces during the vertical jumps. The jump height (cm) is calculated from the recorded force-time history using the software provided by the plate manufacturer.

* The traditional mobility measurements were performed according to the internationally recognized Neutral 0 method [23,24]. They consider movements in the articulation genu, i.e., between femur, tibia, and patella, and are based on certain reference points. These were uniformly defined in the following way. Axis of rotation: lateral femoral epicondyle; Proximal arm: most prominent aspect of the greater trochanter of the femur; Distal arm: external malleolus of the ankle joint.

**Table 2 ijerph-19-16581-t002:** Parameters assessed for the eight exercises considered.

Exercise	Parameter to Be Measured	
	Sensor-Based Assessment	Traditional Assessment
Passive flexion of the lower leg (knee joint)	Tilt angle of the leg in degrees	
Passive extension	Tilt angle of the leg in degrees	
Active flexion of the lower leg (knee joint)	Tilt angle of the leg in degrees	
Angle reproduction	Deviation from prespecified angle in degrees	
One leg squat	Inward tilt of the lower leg in degrees	Ordinal assessment scale
Drop jump	Inward tilt of the lower leg in degrees	Ordinal assessment scale
Side hops (30 s)	Number of correct jumps	
Vertical jump (on one leg)	Jump height (in cm)	

**Table 3 ijerph-19-16581-t003:** Patient characteristics.

Gender (n = 33)	
female	17 (51.5%)
male	16 (48.5%)
Injured side (n = 33)	
left	14 (42.4%)
right	19 (57.6%)
Type of arthroscopic surgery (n = 33)	
partial meniscectomy	18 (54.5%)
ACL reconstruction cartilage smoothing others	10 (30.3%) 3 (9.1%) 2 (6.1%)
Health problems at contralateral side (n = 33)	
no	24 (72.7%)
yes	9 (27.3%)
Age (n = 33)	
mean (sd)	43.0 (15.4)
range	20–70
BMI (n = 33)	
mean (sd)	24.7 (4.2)
range	19.6–34.0
Time since surgery (months) (n = 33)	
mean (sd)	9.8 (2.8)
range	5–16

**Table 4 ijerph-19-16581-t004:** Characteristics of the measured parameters.

		Traditional Assessment		Sensor-Based Assessment	
		Injured Leg	Contralateral Leg	Injured Leg	Contralateral Leg
Passive flexion	n	66	66	65	66
	mean	134.42	136.74	119.22	121.71
	range	117–149	122–150	90–141	81–146
Passive extension	n	66	66	66	66
	mean	2.94	2.71	−22.36	−22.24
	range	−10–15	−10–15	−31–−7	−32–−10
Active flexion	n	66	66	66	66
	mean	126.95	128.17	101.56	104.24
	range	110–138	115–138	75–122	83–131
Angle reproduction	n	66	66	66	66
	mean	4.47	7.26	4.51	6.44
	range	−15–25	−10–20	−16.82–29.72	−13.47–23.98
One leg squat	n	62	62	62	62
	mean	1.50	1.45	8.83	9.14
	range	0–3	0–3	0.31–27.61	0.8–23.19
Drop jump	n	34	34	34	34
	mean	1.12	1.03	0.67	0.46
	range	0–3	0–3	0–10.48	0–7.76
Side hops	n	63	63	63	63
	mean	29.02	32.44	29.19	32.73
	range	4–68	10–66	4–69	10–65
Vertical jump	n	63	64	64	64
	mean	10.36	11.78	10.07	11.82
	range	3.85–21.33	5.57–20.6	4–21.3	4.9–20.9

**Table 5 ijerph-19-16581-t005:** Reproducibility of raw measurements and side differences assessed by ICC values.

	Raw Measurements		Side Differences	
	Traditional Assessment	Sensor-Based Assessment	Traditional Assessment	Sensor-Based Assessment
Passive flexion	0.88	0.89	0.55	0.55
Passive extension	0.97	0.77	0.91	0.51
Active flexion	0.88	0.91	0.63	0.61
Angle reproduction	0.34	0.27	0.00	0.00
One leg squat	0.64	0.54	0.01	0.25
Drop jump	0.94	0.76	0.29	0.29
Side hops	0.88	0.90	0.48	0.49
Vertical jump	0.93	0.91	0.71	0.76

**Table 6 ijerph-19-16581-t006:** Agreement and association of raw measurements and side differences assessed by ICC values and Pearson correlation, respectively.

	ICC	
	Raw measurements	Side differences
Passive flexion	0.00	0.24
Passive extension	0.00	0.43
Active flexion	0.00	0.30
Angle reproduction	0.91	0.90
Side hops	0.96	0.78
Vertical jump	0.97	0.81
	Pearson correlation	
	Raw measurements	Side differences
Passive flexion	0.48	0.34
Passive extension	0.29	0.42
Active flexion	0.47	0.41
Angle reproduction	0.92	0.91
One leg squat	0.37	0.08
Drop jump	0.25	0.26
Side hops	0.96	0.78
Vertical jump	0.97	0.81

## Data Availability

The data presented in this study are available in the Appendix A.

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
