# Peer review of "Comparing a Sensor for Movement Assessment with Traditional Physiotherapeutic Assessment Methods in Patients after Knee Surgery—A Method Comparison and Reproducibility Study"

_ijerph, 2022, doi:10.3390/ijerph192416581_

Round 1
Reviewer 1 Report
Dear Authors:
First of all, congratulate you for the research developed. It is an interesting work, well developed and of interest to professionals.
ABSTRACT: The study design should be clearly defined. It should convey the main quantitative and significant results.
INTRODUCTION: Correct but could be improved by including current references of great relevance to the object of study (accelerometers as a method of assessment: https://doi.org/10.3390/jcm10010137)
MATERIAL AND METHODS: The lack of detailed explanation of the statistical analysis to be performed at the end of this section is alarming. A calculation of the statistical power of the sample size included is necessary. In addition, the effect sizes of the included statisticians should be included.
RESULTS: Is the study variable analyzed by the authors sex or gender?
Zeros as the last decimal place do not mean anything. Please remove them.
DISCUSSION: The use of so many subsections in this section is not recommended.
The lack of bibliographic references in this section is of concern. Please take note of the recommendations indicated in the Introduction as they may be useful for this section as well.
The format of citations throughout the text is not correct. The authors should pay special attention to correcting this detail.
Kind regards.
Author Response
Dear Authors:
First of all, congratulate you for the research developed. It is an interesting work, well developed and of interest to professionals.
Thanks for this kind evaluation. This is encouraging!
ABSTRACT: The study design should be clearly defined. It should convey the main quantitative and significant results.
The abstract has been changed and provides now more information. The study design is explicitly mentioned and some numerical results are given. Both the implications for the specific sensor as well as the implications for the field in general are mentioned.
INTRODUCTION: Correct but could be improved by including current references of great relevance to the object of study (accelerometers as a method of assessment: https://doi.org/10.3390/jcm10010137)
We agree with the reviewer that there is today a broad literature on sensors and accelerometers. For this reason, we refer in the introduction to overview papers and reviews, and not to single research papers such as the one mentioned.
MATERIAL AND METHODS: The lack of detailed explanation of the statistical analysis to be performed at the end of this section is alarming. A calculation of the statistical power of the sample size included is necessary. In addition, the effect sizes of the included statisticians should be included.
We are a little but puzzled by this comment, as there is already a half page paragraph about the statistical analysis.
We added considerations about the sample size. They are based on aiming at a specific precision of the ICC values. Effect sizes are not presented, as the ICC values are not subjected to any statistical test.
RESULTS: Is the study variable analyzed by the authors sex or gender?
Subgroup analyses were not performed due to the limited sample size.
Zeros as the last decimal place do not mean anything. Please remove them.
We suppose that this refers to the ranges reported for the integer variables age and time since surgery. In these representations, the decimal 0 was removed.
DISCUSSION: The use of so many subsections in this section is not recommended.
We agree with the reviewer that the number of subsections is high and unusual. However, this is simply a consequence of investigating simultaneously eight movement assessments. As the results were very differing across the different assessment types, the discussion required to discuss five separate groups of parameters. Moreover, we felt it was important to distinguish between implications for this specific sensor and implications for the use of sensor-based assessments in general.
The lack of bibliographic references in this section is of concern. Please take note of the recommendations indicated in the Introduction as they may be useful for this section as well.
There are already eleven references in the discussion, putting the results for the single modalities into the context of the current knowledge about reproducibility. We agree that there are no further new references with respect to the two sections about implications. This is the simple consequence of the fact that – to the best of our knowledge – this is the first publication comparing a sensor-based assessment with a traditional physiotherapeutic assessment. We made this point now clearer by relating our results to the recommendations developed by Priel et al. based on their recent systematic review.
The format of citations throughout the text is not correct. The authors should pay special attention to correcting this detail.
The format of the citations is corrected.
Reviewer 2 Report
Wearable sensors offer the opportunity for patients to perform a self-assessment of their function concerning various movement exercises. Complementary commercial products have the potential to change the communication between patients and physiotherapists during the recovery process. However, even if they turn out to be user-friendly, there remains the question of to which degree the results are reliable and comparable with those obtained by traditional assessment methods.
The paper presents a study comparison between wearable sensors-based assessment and traditional physiotherapeutic assessment in patients after knee surgery.
In particular, thirty-three patients recovering after arthroscopic knee surgery performed eight movement exercises, and corresponding parameters were assessed using a sensor-based and traditional biomechanical assessment.
The whole assessment procedure was repeated, and the intra-class correlation coefficient quantified reproducibility and agreement.
The treated argument is of central interest in electronics, computer science, and the medical field. Nevertheless, the paper presents the following weaknesses:
- The study's outcome is unclear, and it is unclear if such an outcome is innovative or not.
- It is unclear what type of technology is used in traditional assessments.
- It is tough to visualize the performed activities from the provided description. Therefore, the authors should provide some video recordings of both wearable sensors-based and traditional assessments.
- The Abstract and Introduction should clearly state the achievements/conclusions of the work.
- The paper gives the idea that a methodology is missing and everything the authors do concerns data collection and comparison between wearable-based and traditional assessments.
- Moreover, it is unclear if the collected data have been preprocessed concerning noise removal, missing data, etc.
Author Response
The treated argument is of central interest in electronics, computer science, and the medical field.
We are glad about this positive comment.
Nevertheless, the paper presents the following weaknesses:
- The study's outcome is unclear, and it is unclear if such an outcome is innovative or not.
We tried to be more specific about the aim and objective in the abstract and introduction. The innovative aspect is emphasized in the abstract by mentioned that this study investigates a recently developed and commercially available sensor.
- It is unclear what type of technology is used in traditional assessments.
The technology in the traditional assessments was varying. Partially a goniometer was used, partial a visual assessment, partially manual counts or a force plate. This variation is visible in Table 1, but it is now mentioned more explicitly in the manuscript.
- It is tough to visualize the performed activities from the provided description. Therefore, the authors should provide some video recordings of both wearable sensors-based and traditional assessments.
Within the limited time frame of 10 days for the revision, it was not possible to produce such videos. However, we provide a link to videos provided by the manufacturer of the sensor. We offered the opportunity to add further videos to the editor.
- The Abstract and Introduction should clearly state the achievements/conclusions of the work.
The Abstract was revised accordingly.
- The paper gives the idea that a methodology is missing and everything the authors do concerns data collection and comparison between wearable-based and traditional assessments.
We are not sure about the intention of this comment. We agree with the reviewer that the paper points to a fundamental issue with respect to the emerging use of sensor-based technologies starting to replace traditional assessment methods. This step requires a careful comparison between traditional approaches and sensor-based approaches in order to avoid misinterpretations. In principle, the methodology for such comparisons is in place, as indicated by this study. A different story is the initiation of (regulatory) processes in order to ensure that sensors are only entering the market after such comparisons have been made. The new regulation for medical devices introduced by the EU might be helpful here.
- Moreover, it is unclear if the collected data have been preprocessed concerning noise removal, missing data, etc.
There was no pre-processing as part of the study. This is now mentioned explicitly in the manuscript. Of course, there is some data pre-processing performed by the software of the sensor. We tried to emphasize more clearly that the output of the sensor for each exercise is just one single number.
Reviewer 3 Report
It doesn't feel like creative research, it feels like a regular research report. The work is not innovative either in theory or in technical solutions.
Author Response
It doesn't feel like creative research, it feels like a regular research report. The work is not innovative either in theory or in technical solutions.
We agree with the reviewer that the work is neither innovative with respect to measurement theory nor technical solutions. There is no attempt to improve measurement procedures.
However, the paper is innovative from the perspectives of health services research and health technology assessment. Several reviews have required comparisons of this type.
Reviewer 4 Report
Review of the manuscript titled Comparing a sensor for movement assessment with traditional physiotherapeutic assessment methods in patients after knee surgery – a method comparison and reproducibility study
First, the time and effort put into preparing the manuscript by the authors should be appreciated. However, some major and minor issues need to be addressed.
General comments
My most significant concern regarding the present study is its aim and methodology. Firstly, the objective needs to be revised and rewritten. The purpose of an experimental scientific investigation cannot be to inform potential users of some sensors about the difference they expect when compared to traditional assessment techniques. Please determine an aim suitable for a scientific study. Also, I wouldn't say that some methods are traditional. They are standards, not tradition. Secondly, the methods shouldn't be compared as totally different tests were carried out for the two. Suppose the authors wanted to validate the sensors and the tests provided in the novel system. In that case, they should perform the tests using sensors and validate the values using a standard goniometer. On the other hand, they could complete the tests according to standard methodology using standard goniometers and sensors.
The methodology is not clearly presented. Please add photos of the tests and how to measurements were carried out.
Specific comments
Line 18: I wouldn't consider the assessment of a range of motion as a traditional biomechanical assessment. Please revise this sentence.
Lines 19-20: As long as based on the calculated Intraclass Correlation Coefficient, ICC, we can assess the reliability of a device or measurement protocol, the assessment of agreement of two methods needs another statistical method. Please consult it with the statistician. Also, in my opinion, the agreement cannot be mentioned in this manuscript as, using the two devices, totally different tests were made, so we could predict that, in some cases, the results will be different. For example, when it comes to the passive flexion of the lower leg, even the position (sitting vs. lying flat in a supine position) provides different results, not to mention that in the case of sensor-based assessment, it was a participant who performed the movement in the examined limb. At the same time, according to the standard methodology, it was the examiner.
Line 40: "Corresponding products tend to enter the market" – please revise and rewrite this sentence.
Line 42: The results of what?
Lines 55-56: Again, the results of what?
Line 58: I don't understand what the authors mean by physiotherapeutic management of the rehabilitation process.
Line 59: I disagree with the statement that patients can just present the results of their home assessments to the physiotherapist instead of performing the movement exercise under the supervision of the physiotherapist. It's clear to me that, for sure, this kind of application can support partially supervised home-based physiotherapy. Also, it can help monitor the progress of physiotherapy. It can also be helpful when seeing a physiotherapist is often impossible. We could see those situations, for example, during the pandemic. Or sometimes it might be because of the long distance to a physiotherapist. However, I disagree that this is so easy that patients can make a self-assessment and just present the results to the physiotherapists. Please think again about the possible areas where the application can be helpful or be more precise in your statements.
Author Response
First, the time and effort put into preparing the manuscript by the authors should be appreciated.
Thanks for this kind evaluation. This is encouraging!
However, some major and minor issues need to be addressed.
General comments
My most significant concern regarding the present study is its aim and methodology. Firstly, the objective needs to be revised and rewritten. The purpose of an experimental scientific investigation cannot be to inform potential users of some sensors about the difference they expect when compared to traditional assessment techniques. Please determine an aim suitable for a scientific study. Also, I wouldn't say that some methods are traditional. They are standards, not tradition. Secondly, the methods shouldn't be compared as totally different tests were carried out for the two. Suppose the authors wanted to validate the sensors and the tests provided in the novel system. In that case, they should perform the tests using sensors and validate the values using a standard goniometer. On the other hand, they could complete the tests according to standard methodology using standard goniometers and sensors.
We tried to be clearer about the concrete objective of this investigation and distinguish between the objective of the study and the motivation for the study. Moreover, we name now clearly the design of the study.
We used the term “traditional” to emphasize that there have been methods to assess the parameters of interest for a long time and that clinicians and physiotherapists are used to interpret the results of these assessments. Of course, we agree with the reviewer that these are “standard methods” reflecting the current practice. However, the essential aspect for us is that they reflect what has been done so far in contrast to the “modern” methods provided by sensors. Hence, we feel that “traditional” as a contrast to “modern” is here highly adequate.
However, we try to motivate the use of this term in the introduction by emphasizing that the methods have been used traditionally. We also mention that they reflect current standards.
We also agree with the reviewer that some of the observed disagreements are not surprising, as there are conceptual differences in the measurement procedures. However, the point is that in spite of these differences, the manufacturer of the sensor used terms such as “active flexion” or “passive extension”, suggesting that they claim to be able to measure the same as with traditional assessments using the same term. Hence in our opinion it is important to point out that using the same phrases does not necessarily mean to assess the same quantity. In addition, the (conceptual) differences do not exclude that the procedures result in agreeing (or at least highly correlating) numbers. Hence there is still a need for the investigation presented in this paper.
We agree with the reviewer that a validation of the sensor would require to use a goniometer while performing exactly the same test. However, such a validation is not the aim of this paper. Such a validation has been already performed by Mitternacht et al, using even more precise measurement procedures.
The methodology is not clearly presented. Please add photos of the tests and how to measurements were carried out.
Table 1 is now supplemented with a link to videos provided by the manufacturer of the sensor. We also offered to the editor to create some videos by our own. However, this was not possible within the time frame of 10 days.
Specific comments
Line 18: I wouldn't consider the assessment of a range of motion as a traditional biomechanical assessment. Please revise this sentence.
See the comment above. The phrasing in the abstract has been adjusted accordingly.
Lines 19-20: As long as based on the calculated Intraclass Correlation Coefficient, ICC, we can assess the reliability of a device or measurement protocol, the assessment of agreement of two methods needs another statistical method. Please consult it with the statistician.
We agree with the reviewer that in assessing the agreement of methods there are alternative approaches which are usually preferred. These approaches (such as the limits of agreement) are using the observed differences and provide numbers directly interpretable in terms of the magnitude of the differences to be expected. The ICC plays usually only a secondary role, as the number produced has no direct connection to the differences. Moreover, the ICC depends on the range of the variable of interest observed and hence on the specific patient population considered. However, the ICC has the advantage to produce a number interpretable across different studies focusing on variables with different ranges of possible values.
This is exactly the situation given in this study. The value ranges of the eight different variables are highly varying, as visible in Table 4. As the aim of the investigation is to differentiate between the eight parameters with respect to the value of the new sensor, it seems to be wise and necessary to focus on numbers allowing such a comparison. Consequently, the ICC was chosen as the main parameter of interest. The population dependence of the ICC values plays here a minor role, as all eight parameters are evaluated in the same population.
Of course, readers interesting in a specific exercise and parameter may prefer to see a standard analysis to judge the agreement between the two modalities with respect to one parameter in clinically meaningful units. For this reason, such standard analyses are provided in the additional material.
We did not consult a further statistician. One of the co-authors (WV) has worked on the methodology of agreement studies for many years (e.g. Vach W. The dependence of Cohen's kappa on the prevalence does not matter. J Clin Epidemiol. 2005 Jul;58(7):655-61. doi: 10.1016/j.jclinepi.2004.02.021. Vach W, Gerke O. How Replicates Can Inform Potential Users of a Measurement Procedure about Measurement Error: Basic Concepts and Methods. Diagnostics (Basel). 2021 Jan 22;11(2):162. doi: 10.3390/diagnostics11020162.) and performed the statistical analyses of many reproducibility and agreement studies.
Also, in my opinion, the agreement cannot be mentioned in this manuscript as, using the two devices, totally different tests were made, so we could predict that, in some cases, the results will be different. For example, when it comes to the passive flexion of the lower leg, even the position (sitting vs. lying flat in a supine position) provides different results, not to mention that in the case of sensor-based assessment, it was a participant who performed the movement in the examined limb. At the same time, according to the standard methodology, it was the examiner.
As pointed out above, we agree with the reviewer that the low agreement is in some cases not unexpected. However, there should be in this case some correlation. Even this is not the case. We think this is worth to be pointed out.
Line 40: "Corresponding products tend to enter the market" – please revise and rewrite this sentence.
The sentences read now:
«If function is related to specific movement patterns, wearable sensors are promising to allow a simple and objective assessment and may replace existing biomechanical assessment methods [9,10]. Such sensors are becoming increasingly commercially available.»
Line 42: The results of what?
The sentence reads now
«Even if they turn out to be user-friendly, they face clinicians and patients with the question to which degree the numerical results produced by the sensors are reliable and comparable with those obtained by assessment methods traditionally used.»
Lines 55-56: Again, the results of what?
The sentence reads now
«The output from the app consist of a single number per exercise and can be tracked over time in the case of repeated applications.»
Line 58: I don't understand what the authors mean by physiotherapeutic management of the rehabilitation process.
We have rephrased this passage and focus now on the communication between the physiotherapist and the patient.
Line 59: I disagree with the statement that patients can just present the results of their home assessments to the physiotherapist instead of performing the movement exercise under the supervision of the physiotherapist. It's clear to me that, for sure, this kind of application can support partially supervised home-based physiotherapy. Also, it can help monitor the progress of physiotherapy. It can also be helpful when seeing a physiotherapist is often impossible. We could see those situations, for example, during the pandemic. Or sometimes it might be because of the long distance to a physiotherapist. However, I disagree that this is so easy that patients can make a self-assessment and just present the results to the physiotherapists. Please think again about the possible areas where the application can be helpful or be more precise in your statements.
We agree with the reviewer that there are many ways in which the sensor can be used, and this is now mentioned explicitly. However, for some of the areas of applications it is not so much essential that the measurements of the sensor are comparable with traditional assessments. For example, if the patient is monitoring his or her progress over time during rehabilitation, it is sufficient that the values produced by the sensor are a valid representation of the functional status. The values do not have to be numerically similar to traditional assessments.
The direct comparability of the values is essential, if a single assessment previously performed by a physiotherapist is replaced by a sensor-based assessment. For this reason, we focus in the discussion of the potential application scenarios on such a “replacement” situation. The scenario of replacing an assessment by the physiotherapist with presenting the results of a home-made assessment seems to us also a rather relevant one, as it would be attractive for the physiotherapists to have more time for the patients. However, we agree with the reviewer that this is not an easy step.
The whole passage reads now
«The system is (also) intended to be used by patients at home supporting various forms of self-assessment. This way, it can also become part of the communication and interaction between the physiotherapist and the patient. In particular, patients may just present the results of their home assessments to the physiotherapist instead of performing the movement exercise under the supervision of the physiotherapist. In this situation, the physiotherapist must be able to judge the reliability and validity of the output produced by the sensor compared to the assessment methods previously used in his or her routine. The same holds for clinicians used to interpret results of assessments performed by physiotherapists today.»
Round 2
Reviewer 1 Report
Dear Authors,
First of all, I would like to congratulate you for the work done in the correction of the manuscript. It has been an honour for me to participate in its evaluation and improvement.
Regarding the work carried out by the authors to correct the manuscript, I consider that they have developed a detailed and effective work to improve the scientific and formal quality of the manuscript.
Therefore, I now consider that the manuscript does meet the scientific and formal requirements for publication in this Journal.
Kind regards.
Author Response
First of all, I would like to congratulate you for the work done in the correction of the manuscript. It has been an honour for me to participate in its evaluation and improvement.
Regarding the work carried out by the authors to correct the manuscript, I consider that they have developed a detailed and effective work to improve the scientific and formal quality of the manuscript.
Therefore, I now consider that the manuscript does meet the scientific and formal requirements for publication in this Journal.
Kind regards.
We are very happy about this judgement and thank the reviewer for all efforts.
Reviewer 2 Report
The authors provided an improved version of their manuscript based on the received comments. Nevertheless, again in the current form it is not clear which are the main achievement they obtain.
They compare commercial sensors with the traditional assessment method. However, it is not clear the final result. This type of comparison has already been under focus from different studies [1-5].
[1] Guo, Christine C., Patrizia Andrea Chiesa, Carl de Moor, Mir Sohail Fazeli, Thomas Schofield, Kimberly Hofer, Shibeshih Belachew, and Alf Scotland. "Digital Devices for Assessing Motor Functions in Mobility-Impaired and Healthy Populations: Systematic Literature Review." Journal of Medical Internet Research 24, no. 11 (2022): e37683.
[2] Del Din, Silvia, Cameron Kirk, Alison J. Yarnall, Lynn Rochester, and Jeffrey M. Hausdorff. "Body-worn sensors for remote monitoring of parkinson’s disease motor symptoms: vision, state of the art, and challenges ahead." Journal of Parkinson's disease 11, no. s1 (2021): S35-S47.
[3] Ossig, Christiana, Angelo Antonini, Carsten Buhmann, Joseph Classen, Ilona Csoti, Björn Falkenburger, Michael Schwarz, Jürgen Winkler, and Alexander Storch. "Wearable sensor-based objective assessment of motor symptoms in Parkinson’s disease." Journal of neural transmission 123, no. 1 (2016): 57-64.
[4] Eskofier, Bjoern M., Sunghoon I. Lee, Jean-Francois Daneault, Fatemeh N. Golabchi, Gabriela Ferreira-Carvalho, Gloria Vergara-Diaz, Stefano Sapienza et al. "Recent machine learning advancements in sensor-based mobility analysis: Deep learning for Parkinson's disease assessment." In 2016 38th Annual International Conference of the IEEE Engineering in Medicine and Biology Society (EMBC), pp. 655-658. IEEE, 2016.
[5] Assessing motor mobility with sensors, https://scholar.google.it/scholar?hl=en&as_sdt=0,5&q=assessing+motor+mobility+with+sensors
Author Response
The authors provided an improved version of their manuscript based on the received comments. Nevertheless, again in the current form it is not clear which are the main achievement they obtain.
We are sorry about that this is still not clear.
The papers the reviewers suggested as references have been studied by us. They refer to the use of sensors for motor function assessment in patients with or suspected for Parkinson disease or related diseases. The typical validation of sensor-based assessments is described in the review by Guo et al in the following way:
"The most common validity criterion was clinical condition(37/87, 43%), which was used in many of these studies to establish known-group construct discriminant validity of sensor-derived motion data by comparing participants with a diseased condition to healthy controls. The second most common validity criterion was the clinical validity established by assessing the convergence or concurrence with traditional standardized clinical assessments (30/87, 34%). Other criteria were clinician ratings (7/87, 8%), research device (9/87, 10%), treatment status (3/87, 3%), and patient-reported outcome (1/87, 1%)."
This description demonstrates that it is usual in this field to validate sensors against a clinical assessment or a clinical diagnosis.
The situation is different in the field of movement assessment in orthopaedic patients. Movement assessments aim at measuring a single numerical parameter such as an angle, a jump height, or a number of jumps. In principle, it is possible to measure these numbers in an objective and precise manner, for example via multi camera video recordings. Hence it is possible to conduct a (non-clinical) validation by comparing the output from the sensor with the true values.
This is the type of validation usually done, as described for example in the systematic review by Prill et al (2021). Such a validation has been performed for the sensor considered in our paper, too (Mitternacht et al 2022). However, this does not imply that the sensor is also clinically useful (as also pointed out by Prill et al). The point is, that the parameters measured need not to coincide with those considered in traditional assessment procedures. This point is nicely illustrated by the results of our paper. Active flexion, passive flexion, and passive extensions are well established clinical constructs, and the study by Mitternacht et al suggests that the sensor-based values are close to the truth. However, the sensor-based measurements use another exercise set-up than the traditional assessment procedures, so the latter refer to a different truth. This does not need to matter, if in both cases the truth is reflecting the same clinical construct (as suggested by using identical names). Our investigation reveals that this has to be questioned, as even the correlation between the values generated by the sensor and the values generated by the traditional assessment is rather low. With respect to knee-instability, the situation is even worse with nearly no association.
This does not mean that the sensor is invalid. It just implies that further steps are necessary to demonstrate the clinical value of the measurements provided by the sensor.
The evaluation of sensors for assessing motor function in mobility-impaired subjects could benefit from a direct comparison with clinical diagnoses and clinical assessments, such that it is already now possible to judge their clinical usefulness to some degree. The assessment of movement parameters in orthopaedic patients is not so far. The essential step to prove clinical utility has still to be made. Our paper tries to contribute to this step by informing about potential differences to a traditional assessment. This depicts the current utility of the sensor if it is used in a situation replacing the traditional assessment.
We hope that this clarifies your concerns.
They compare commercial sensors with the traditional assessment method. However, it is not clear the final result. This type of comparison has already been under focus from different studies [1-5].
As pointed out above, we agree with the reviewer that direct comparisons with clinical assessments have been made in the field of assessing motor function. Although at first sight there seems to be some similarity between assessment of motor function and assessment of movement parameters, the actual overlap seems to be small. Guo et al present the following list of motor outcomes:
"Peak upper limb velocity, Upper limb velocity, Spiral tracing, Depressive tendencies, Finger tap speed, Flight time, Hold time, Bradykinesia score, Dyskinesia score, Spiral tracing, Correct finger taps, Finger tap accuracy, Finger tap count, Finger tap duration, Finger tap interval, Finger tap reaction time, Finger tap rhythm, Finger tapping test, Flight time, Hold time, Joint velocity, Step cadence, Step count, Step length, Stride duration, Turning speed, Walking speed, Lower limb velocity, Joint velocity, Rest tremor, Trunk acceleration, Stroop Color and Word Test, Chorea score, Finger tap speed, Step cadence, Spiral tracing"
None of the outcomes considered in this list is assessed by the sensor considered in our paper.
Reviewer 3 Report
Compared with the original manuscript, there are no substantial changes, therefore, it is recommended to reject the draft.
Author Response
Compared with the original manuscript, there are no substantial changes, therefore, it is recommended to reject the draft.
We are sorry about this judgement.
Please note that other reviewers have been very satisfied with the general set up of the paper. For this reason we tried to take your concerns into account without making substantial changes.
Reviewer 4 Report
All of my previous comments were addressed. In my opinion, the manuscript has been highly improved.
Author Response
All of my previous comments were addressed. In my opinion, the manuscript has been highly improved.
We are grateful for this positive evaluation and thank the reviewer for all efforts.